# Coating Seeds with *Paenibacillus polymyxa* ZF129 Microcapsule Suspension Enhanced Control Effect on *Fusarium* Root Rot and Promoted Seedling Growth in Cucumber

**DOI:** 10.3390/biology14040375

**Published:** 2025-04-05

**Authors:** Jiayi Ma, Jialin Liu, Yanxia Shi, Xuewen Xie, Ali Chai, Sheng Xiang, Xianhua Sun, Lei Li, Baoju Li, Tengfei Fan

**Affiliations:** State Key Laboratory of Vegetable Biobreeding, Institute of Vegetables and Flowers, Chinese Academy of Agricultural Sciences, Beijing 100081, China; 82101201141@caas.cn (J.M.); liujialin199806@163.com (J.L.);

**Keywords:** *Paenibacillus polymyxa* ZF129, cucumber *Fusarium* root rot, microcapsule suspension, biological control

## Abstract

In this study, a *Paenibacillus polymyxa* ZF129 microcapsule suspension (CS-ZF129) was prepared. *P. polymyxa* ZF129 was microencapsulated by gum arabic via the spray-drying method. Coating seeds with the CS-ZF129 not only increased the control efficacy against cucumber root rot to 46.30 ± 0.02%, but also promoted the growth of the cucumber roots and other aspects, effectively improving the colonization ability of *polymyxa* ZF129 in the cucumber root system, which was analyzed to clarify the biological control mechanism of the cucumber *Fusarium* root rot disease through the enzyme activity. This study provides an efficient means based on seed coating for the biological control of cucumber *Fusarium* root rot.

## 1. Introduction

The cucumber is an important vegetable worldwide. *Fusarium* root rot, a soil-borne disease caused by *Fusarium solani*, represents a considerable threat to the cultivation and production of cucumbers [1]. *Fusarium solani* primarily inhabits the soil and has the capacity to survive harsh external conditions through the formation of numerous chlamydospores, which are capable of enduring for more than three years [2]. During the development of the seedling, distinctive spots emerge on the cotyledons and stem base. In severe cases, this can result in the death of the plant [3]. As the disease progresses to the adult stage, water-soaked spots develop at the base of the cucumber stem, impeding the absorption of water and nutrients. Under favorable environmental conditions, the disease can rapidly lead to the death of cucumber plants within 3–5 days.

While chemical pesticides remain the conventional approach for disease management, their indiscriminate application has prompted critical reassessment due to their escalating operational expenditures and ecological contamination risks. Concurrently, prolonged pesticide utilization accelerates pathogen resistance development through both the target-site modification and detoxification enhancement mechanisms, systematically diminishing the long-term viability of the current control paradigms [4]. The fungicides propiconazole and difenoconazole and premixtures of pydiflumetofen + fludioxonil or pydiflumetofen + difenoconazole provided excellent root decay control. Azoxystrobin provided excellent (69.9%) control of the *Phoma* root decay of table beets in 2019, and lesser (40%) control in 2021 [5]. Biological control represents a sustainable alternative that circumvents the challenges posed by chemical residues. In particular, the process of coating seeds with biologically derived control bacteria represents an effective and labor-saving approach for the management of soil-borne diseases. Nevertheless, the extensive deployment of microbial seed coating remains constrained by the persistent challenge of guaranteeing the survival of the biocontrol bacteria under a multitude of biotic and abiotic stress conditions. The formulation of microbial seed treatments typically comprises three fundamental components: microorganisms, carriers, and additional additives. The formulation of the microbial seed treatment significantly influences the microbial survival throughout processing, storage, and application, as well as the effectiveness and economic viability of their application on crops. For example, Georgakopoulos et al. (2002) developed a *Pseudomonas* seed treatment utilizing peat as a carrier material to combat wilt in sugar beets and cucumbers. This demonstrated the ability of *Pseudomonas aeruginosa* to endure for more than two years at room temperature [6]. Similarly, Głodowska et al. (2017) developed a seed-coating formulation incorporating biochar as a carrier, which was demonstrated to sustain a high rhizobial population for four months, thereby ensuring the efficient nodulation of soybean root systems [7]. Colla et al. (2015) [8] coated wheat seeds with endophytic fungi, increasing the stems, root biomass, and leaf numbers of the winter wheat seedlings by 23%, 64%, and 29%, respectively. The yield increased by 8.3% to 32.1%, the grain protein concentration increased by 6.3%, and the grain potassium, phosphorus, iron, and zinc concentrations also generally increased [8]. Angelopoulou et al. (2014) found that the use of two seed coating agents, Bacillus subtilis K165 and the non-pathogenic *Fusarium oxysporum* F2, delayed the onset of eggplant wilt disease and significantly reduced the disease index, and also promoted plant growth [9].

*Paenibacillus polymyxa* is an important microbe that is widely studied for its biocontrol abilities. *P. polymyxa* can produce a variety of antimicrobial substances [10], such as polymyxin, polypeptidin, fusarium, jolipeptin, paenilan, gavaserinsal, and stavalin [11], which can control a variety of plant diseases, such as wheat sheath blight [12], cucumber *Fusarium* root rot [13], and Pitaya canker [14]. *P. polymyxa* is also an important plant growth-promoting rhizobacteria. It directly or indirectly promotes plant growth by producing plant hormones, siderophores, phosphate-solubilizing enzymes, nitrogenases, and so on [15]. It has been reported that *P. polymyxa* isolated from the barley rhizosheath is IAA-producing and promotes the formation of the barley rhizosheath, increasing the barley yield [16]. In addition, Hongbo Yuan reported that *P. polymyxa* NI4 can produce protease, cellulase, and β-1,3-glucanase, which affect the expression of membrane- and energy metabolism-related genes in pathogenic bacteria and effectively reduce the incidence of Pear Valsa canker occurrence [17]. J2–4, which was isolated from ginger *P. polymyxa* J2–4, showed promising results against southern root-knot nematodes. J2–4 inhibited nematode development by inducing resistance in cucumber roots through signaling to salicylic acid and jasmonic acid, resulting in a 65.94% reduction in galls and a 51.64% reduction in eggs in a pot experiment [18]. The ZF129 genome consists of one 5,703,931 bp circular chromosome and two 79,020 bp and 37,602 bp plasmids, designated pAP1 and pAP2, respectively. *P. polymyxa* ZF129 is largely mediated by phytohormone production, increased nutrient availability, and biocontrol mechanisms; the inhibition rate of *P. polymyxa* ZF129 against *F. solani* in a plate confrontation test was 51.37% [19]. Previous studies have shown that ZF129 has a good preventive and therapeutic effect on clubroot disease in Pak Choi after seed packaging [20].

This study developed a novel microbial seed coating agent (MSC-129) for the sustainable management of cucumber *Fusarium* root rot, employing *Paenibacillus polymyxa* ZF129 as the core biocontrol agent. The formulation process incorporated two critical technological innovations: (1) the precise encapsulation of ZF129 spores through spray-drying technology, using gum arabic as the protective matrix, and (2) the strategic suspension of the obtained microcapsules in a corn oil supplement. The control effect of the CS-ZF129 against cucumber *Fusarium* root rot and its growth-promoting effect were investigated, achieving 46.30 ± 0.02% disease suppression while promoting root growth. The colonization behavior of *P. polymyxa* ZF129 in the roots of the cucumber plants was analyzed. Moreover, the rhizosphere soil enzyme activity and the root defense enzyme activity were also determined.

## 2. Materials and Methods

### 2.1. Materials and Bacterial Strains

Sodium carboxymethyl cellulose (Sigma Aldrich, St. Louis, MO, USA). Skim milk (Sanyuan Food Co., Ltd., Beijing, China). Ammonium sulfate, Congo red, (NH_4_)_2_SO_4_, MgSO_4_·7H_2_O, MnSO_4_·H_2_O, FeSO_4_·7H_2_O, NH_4_NO_3_, KH_2_PO_4_, and lecithin (Aladdin Biochemical Technology Co., Ltd., Shanghai, China). Gum arabic, maltodextrin, dextrin glucose, agar, NaCl, KCl, yeast extract, and tryptone (Solarbio Biotechnology Co., Ltd., Beijing, China). All of the reagents were of analytical grade.

Cucumber seeds (*Cucumis sativus* L., cultivar Zhongnong 6) (China Vegetable Seed Technology Co., Ltd., Beijing, China). *P. polymyxa* ZF129 (NCBI Accession: CP040829.1) and *Fusarium solani* (Institute of Vegetables and Flowers, Chinese Academy of Agricultural Sciences, Beijing, China).

The skim milk media included 10.00 g of glucose, 0.50 g of (NH_4_)_2_SO_4,_ 0.50 g of yeast extract, 0.30 g of NaCl, 0.30 g of KCl, 0.30 g of MgSO_4_·7H_2_O, 0.03 g of MnSO_4_·H_2_O, 0.20 g of lecithin, 0.03 g of FeSO_4_·7H_2_O, 15.00 g of agar, and 1 L of distilled water.

Mongina medium: 100 mL of skim milk, 2.00 g of agar, and 100 mL of lysogeny broth medium.

The CMC-Na media included 15.0 g of CMC-Na_,_ 1.00 g of NH_4_NO_3_, 1.0 g of yeast extract, 0.5 g of MgSO_4_·7H_2_O, 1.0 g of KH_2_PO_4_, and 1 L of distilled water.

### 2.2. Plant Materials and Bacterial Strains

The bacterial stocks of the *P. polymyxa* ZF129 used for the inoculation were stored at 20 °C in lysogeny broth (LB) supplemented with 20% (*v*/*v*) glycerol. The *P. polymyxa* ZF129 strain was cultured in the LB at 30 °C with shaking at 180 rpm in the dark for 48 h. The cell concentrations were determined by measuring the absorbance at 660 nm.

### 2.3. Preparation of P. polymyxa ZF129 Microcapsule Powder via Spray-Drying Method

In a separate experiment, a carrier solution comprising 20% gum arabic was prepared by mixing the carrier solution and a suspension of *P. polymyxa* ZF129 at a concentration of 1 × 10^8^ cfu/mL in a 1:1 ratio. A magnetic stirrer (IKA, Staufen, Baden-Württemberg, Germany) was used to maintain continuous stirring of the mixture at 800 rpm, and a peristaltic pump (Cole-Parmer, Vernon Hills, IL, USA) was used to feed the mixture at a rate of 20%; the feed rate was 200 mL/h, the nozzle diameter was 1 mm, the needle (Buchi, Flawil, St. Gallen, Switzerland) was passed through at a rate of 2 s, and the temperature of the air inlet was set at 120 °C.

The *P. polymyxa* ZF129 microcapsule powder was stored at 4 °C and 25 °C. The *P. polymyxa* ZF129 contents in the microcapsule powders prepared under different carrier (gum arabic, maltodextrin, and dextrin) and spray-drying conditions were determined via the dilution coating method, which was repeated three times for each treatment and measured at seven-day intervals. This approach was employed to analyze the optimal conditions for the preparation of the *P. polymyxa* ZF129 microcapsule powder.

### 2.4. Field Emission Scanning Electron Microscopy (FESEM)

The *P. polymyxa* ZF129 microcapsule powder samples were spread onto conductive carbon tape, and then stuck on an SEM stub. Furthermore, the samples were sputter-coated with gold to a thickness of 10 nm (20 mA, 80 s) via a JFC-1600 Auto Fine Coater (JEOL, Tokyo, Japan). To visualize the cells, the freeze-dried and loaded-bacteria macrobeads were step-wise dehydrated in ethanol (series of 50 %, 70 %, 90 %, and 100 %) and then in pure acetone, critical point-dried, and coated with gold. FESEM micrographs were obtained with a JSM-7600F FESEM (JEOL, Tokyo, Japan) at 3 kV and were taken at various magnifications.

### 2.5. Preparation of CS-ZF129 and Seed Coating

Briefly, 10 g of ZF129 microcapsule powder, 10 g of Tween 80, and 0.05 g of blue coloring agent were added to 80.95 g of corn oil. Afterward, the mixture was homogenized by a high-speed shearing machine (T18, IKA, Germany) and sheared at 3000 rpm for 20 min. The obtained *P. polymyxa* ZF129 microcapsule suspension (CS-ZF129) was kept in a serum bottle for further characterization.

The cucumber seeds were coated with the CS-ZF129 via the plate-shaking method. Typically, 10 g or 100 g of cucumber seeds were placed into a 9 cm diameter glass Petri dish. Then, 1 g of the seed coating agent was dropped onto the seeds. The Petri dish was shaken by hand for three minutes to ensure that all of the seeds were evenly coated with the CS-ZF129. The coated seeds were then dried at 25 °C for 2 days.

### 2.6. Storage Stability

The CS-ZF129 was stored at 4 °C, 25 °C, and 54 °C, a relative humidity of 60 ± 5%, and a light intensity of 500 lux, alternating between 12 h of light and 12 h of darkness, and the number of viable bacteria in the CS-ZF129 was determined via the dilution coating method at different time points over 6 months.

The cucumber seeds coated with the CS-ZF129 were stored at 25 °C for 3 months, after which the number of viable bacteria on the surface of the cucumber seeds was determined. One coated cucumber seed was placed in a 2 mL centrifuge tube, and 1 mL of sterile water was added. The mixture was fully vortexed and shaken for 10 min to ensure that the bacteria on the surface of the seed were completely eluted. The treatment mixture was diluted and coated, and the biocontrol bacteria content was measured as each seed’s surface-loaded biocontrol bacteria content.

### 2.7. Effects of CS-ZF129 on Growth Promotion of Cucumbers and Control of Cucumber Root Rot

Both the coated and naked seeds were sown in trays (540 mm × 280 mm) containing 4000 g of seedling substrate. The cucumber seedlings were cultured in a greenhouse under the following conditions: 25 °C, a 16 h photoperiod, and 70% RH. Moreover, the plant heights, shoot weights, and root lengths of all of the tested plants were recorded after 30 days according to Huang et al. [21]. Briefly, the plant height was measured starting from the base of the stem to the highest tip of the leaf via a standard ruler. The shoot weight was measured by weighing all parts of the plant except for the roots with an electronic balance. The length of the roots was measured starting from the base of the root to the tip of the longest root via a standard ruler.

To investigate the potential of the CS-ZF129 against cucumber *Fusarium* root rot, experiments were conducted using either coated seeds or naked seeds. We designed CS-ZF129/seed weight ratios of 1:10 and 1:100, with naked seeds sown in a sterile substrate soil and a pathogen-containing soil used as the non-treated and pathogen controls, respectively. The pathogen-containing soil was prepared by mixing an *F. solani* fermentation broth (10^8^ spores/mL) with the sterilized substrate soil at a ratio of 1:10. The coated and naked cucumber seeds were sown separately in the pathogen-containing soil. Each treatment was replicated three times, with 15 seedlings per replicate.

The infected plants were scored on a 0–4 scale [22]: 0, good plant growth; 1, 1–25% of leaves show yellowing and wilting with slight symptoms on the embryonic axis or cotyledons, but normal growth; 2, 26–50% of leaves wilting and conspicuous necrotic spots on the embryonic axis or cotyledons or the yellowing of one cotyledon, with growth somewhat affected; 3, 51–75% of leaves wither and die, with necrotic spots causing localized wilting, one cotyledon dying, or growth stiffening; and 4, 76–100% leaf death or whole plant death, overall wilting, collapse, and dieback. The disease severity index (DSI) and control effect were calculated via Equations (1) and (2), respectively.DSI = [∑(Number of diseased plants in this grade × disease grade)/(Total number of plants × highest disease grade)] × 100(1)Control effect (%) = [(DSI of infected control − DSI of treatment)/DSI of infected control] × 100(2)

### 2.8. Colonization Ability of P. polymyxa ZF129 in Cucumber Rhizosphere and Its Relationship with Population Changes in F. solani

The colonization of the biocontrol bacteria in the rhizosphere and cucumber roots was studied by using induced rifampicin-resistant ZF129 (ZF129^Rif^). The first step involved the preparation of LB solid media with rifampicin concentrations of 1, 2, 5, 10, 20, 40, 60, 80, and 100 µg/mL. Single colonies of ZF129 were inoculated onto LB plates containing rifampicin at a concentration of 1 ppm, and then, the newly grown single colonies were inoculated onto plates with increasing concentrations of rifampicin (2, 5, 10, etc., up to 100 ppm) [23]. The final growth of the resistant strains was repeated more than three times on plates containing rifampicin at a concentration of 100 ppm to ensure stable resistance.

CS-ZF129^Rif^ was reprinted from the ZF129^Rif,^ and cucumber seeds were coated. The coated and naked seeds were sown in the pathogen-containing soil. On days 1, 3, 5, 10, 20, and 30 after sowing, cucumber root and rhizosphere soil samples were collected. The sampling method was as follows: the cucumber was dug up, the rhizosphere soil and roots were collected, and the rhizosphere soil was sieved through a 2 mm sieve to remove the plant debris. The rhizosphere soil was directly subjected to gradient dilution. The roots were rinsed repeatedly with sterile water and fully ground in a 1.5 mL centrifuge tube for gradient dilution. The treatment mixture was spread on LB plates containing 100 µg/mL of rifampicin, and the biocontrol bacteria contents were converted to the amounts of ZF129^Rif^ per g of sample (CFU/g).

In addition, we used a soil genomic DNA extraction kit (DP336-02) (TIANGEN Biotech, Beijing, China) to extract the rhizosphere soil DNA. This DNA was used as a template for a real-time fluorescence quantitative PCR with F8-1(GCTTCTCCCGAGTCCCA)/F8-2(GCTCAGCGGCTTCCTAT) primers according to the reaction conditions and reaction procedures optimized by Chen Lida et al. [24]. Briefly, the 20 μL reaction volume contained 10 μL of SuperReal PreMix Plus, 0.2 μL of F8-1/F8-2 primer (10 μM), 1 μL of template DNA (210 ng), and 0.4 μL of 50 × ROX Reference Dye. Amplification was performed by using the ABI 7500 Real-Time PCR System (Applied Biosystems, Carlsbad, CA, USA). The fluorescence was detected after each cycle. The cycle threshold (Ct) values were calculated automatically by ABI 7500 software (Applied Biosystems, Carlsbad, CA, USA). Three replicates were set for each treatment, and ddH_2_O was used as a blank control instead of a template. The Ct values of the samples were obtained, and these Ct values were brought to the standard curve of *F. solani* to calculate the contents of *F. solani* in the samples. The obtained data were subsequently converted to the contents of *F. solani* per g of soil sample (CFU/g), with the correlation between ZF129^Rif^ and *F. solani* under investigation.

### 2.9. Determination of Rhizosphere Soil Enzyme Activity and Root Defense Enzyme Activity

Additional root and rhizosphere soil samples collected during the experiment in Section 2.8 were stored until day 30. The root samples were rapidly cooled in liquid nitrogen and stored at −80 °C. The rhizosphere soil samples were air-dried at room temperature, ground, sieved through a 40-mesh sieve, and stored at room temperature. The procedure was carried out according to the instructions of the kit (Beijing Solarbio Science & Technology Co., Ltd., Beijing, China). The activities of peroxidase (POD), superoxide dismutase (SOD), and catalase (CAT) were determined in the cucumber root system. The activities of soil urease (S-UE), soil alkaline phosphatase (S-AKP/ALP), soil saccharase (S-SC), and soil catalase (S-CAT) in the rhizosphere soil were measured, with a reaction temperature of 37 °C and an incubation time of 1 h.

### 2.10. Statistical Analysis

Statistical analyses of all of the data were performed via a one-way analysis of variance (ANOVA) via PASW statistics 21.0. The means of the different treatments for each trait were compared via Duncan’s multiple range test at *p* ≤ 0.05. All means of the different treatments for each trait were acquired from the raw data of the three replications and expressed as the means ± standard deviations (SDs).

## 3. Results

### 3.1. The Morphology of the Microcapsules

The morphology of the *P. polymyxa* ZF129 microcapsules prepared at different magnifications is shown in Figure 1. As illustrated, the *P. polymyxa* ZF129 microcapsules prepared at an inlet air temperature of 120 °C presented a smooth surface, yet the shape became irregular, and concavities were observed. It can be inferred that the formation of the concavities was a consequence of the rapid evaporation of the atomized liquid drops during the spray-drying process [25]. Notably, no fractures or debris were observed on the samples, which indicates that the wall material is capable of withstanding the mechanical forces associated with the expansion and ballooning during spray drying [26].

### 3.2. The Storage Stability of the CS-ZF129 Microcapsules

After mixing the ZF129 microcapsules with the oil solvent and other adjuvants, the ZF129 microcapsule suspension (CS-ZF129) was formed with a red color, as shown in Figure 2. The viable cell count of the CS-ZF129 was investigated at room temperature for 5 weeks. The initial microbial content was 6 × 10^8^ CFU/g; when stored at 25 °C, the biocontrol bacterial content was 3.76 × 10^8^ CFU/mL after 2 weeks of storage, which was 50% lower than the initial biocontrol bacterial content. After 5 weeks of storage, the biocontrol bacterial content ultimately decreased to 8.61 × 10^6^ CFU/mL. The biocontrol bacterial content decreased sharply when the ZF129 cell cultures were stored at the same conditions, and their biocontrol bacterial content decreased from 6 × 10^8^ CFU/mL to 8.06 × 10^6^ CFU/mL, representing a 100-fold reduction in quantity. In the fifth week, the biocontrol bacterial content was 9.9 × 10^2^ CFU/mL. This indicates that microencapsulation can significantly improve the survival ability of *P. Polymyxa* ZF129, effectively improving its shelf life and ensuring that it can better exert its control effects against plant diseases.

### 3.3. Surface Morphology of CS-ZF129-Coated Cucumber Seeds (ZF129m-Seeds)

The surface morphology of the bare cucumber seeds and ZF129m-seeds were imaged by scanning electron microscopy. The bare cucumber seeds showed a multi-layered, tightly arranged fence-like structure with a small amount of debris on the surface. After coating with the CS-ZF129, the surface of the ZF129m-seeds was uniformly covered by microcapsules, with a coating thickness of 21 ± 1.41 μm and a coverage rate of 98.32 ± 0.44% (Figure 3). Meanwhile, some bigger debris also can be observed due to the existence of adjuvants such as emulsifiers and coloring agents.

### 3.4. Growth Response of ZF129m-Seeds in Pot Experiments

A comparison of the growth parameters of the coated and bare cucumber seeds revealed that coating with the CS-ZF129 promoted the growth of the cucumbers, particularly in terms of the root length, fresh weight, and plant height (Table 1). Two types of ZF129m-seeds with different coating agent/seed ratios (1:10 and 1:100) were tested. At 30 days after sowing, the heights of the cucumber plants in T1 (14.52 ± 2.44 cm) were higher than in T2 (11.90 ± 0.88 cm) and CK (11.82 ± 0.84 cm). The root lengths of both T1 (12.38 ± 0.10 cm) and T2 (12.75 ± 0.64 cm) were significantly longer than in CK (8.9 ± 1.08 cm). The shoot weights and root weights of T1 and T2 were also higher than in CK. Notably, T1 showed a heavier root weight than T2, but no difference in shoot weight. This suggests that the growth promotion effect of ZF129 was more obvious in roots than in shoots. The leaf areas and SPDA values were also investigated, but no significant difference was found among the three treatments.

### 3.5. Control Effect of CS-ZF129 Against Cucumber Fusarium Root Rot

The effects of coating the seeds with the CS-ZF129 on the control of cucumber root rot were evaluated in the pot experiments. The results of the pot experiments revealed that the CS-ZF129 treatment was effective in controlling *Fusarium* root rot in the cucumbers (Table 2). The disease index of T3 was the highest, with a value of 68.00 (±1.91). All of the plants presented symptoms of root rot and lodging. The application of T1 and T2 successfully reduced the disease indices to 49.98 (±6.30) and 36.30 (±1.25), respectively, and the control efficacies were 29.01 (±0.09) % and 46.30 (±0.02) %, respectively. Note that we previously reported that if more fungus was present on the seeds, the control would be better, but this did not occur. When we used the 1:100-ratio coating treatment, the protection against cucumber *Fusarium* root rot was lower than that of the 1:10-ratio coating treatment. This may be because the high rate of coating affected the germination. However, we are sure that seed coating with the CS-ZF129 can effectively suppress the occurrence of cucumber *Fusarium* root rot (Figure 4).

### 3.6. Dynamics of Populations of P. polymyxa ZF129^Rif^ and F. solani in Rhizosphere

By measuring the changes in the number of *P. polymyxa* ZF129^Rif^ in the cucumber rhizosphere, we found that *P. polymyxa* ZF129^Rif^ could be stably colonized in both the cucumber root and the rhizosphere soil. Within 20 days after inoculation, the population of *P. polymyxa* ZF129^Rif^ in the cucumber roots and across the different soil layers exhibited an initial increase followed by a decrease, whereas the corresponding *F. solani* population steadily declined and stabilized. A negative correlation was observed between the two (Figure 5).

When the agent/seed weight ratio was 1:10, the number of *F. solani* in the rhizosphere soil gradually decreased from 4.52 × 10^4^ CFU/g to 4.07 × 10^3^ CFU/g from 0 to 10 days, and the numbers of *F. solani* on the first, third, and fifth days were 2.06 × 10^4^ CFU/g, 13.3 × 10^4^ CFU/g, and 1.05 × 10^4^ CFU/g, respectively. Then, the number of *F. solani* in the rhizosphere soil increased slightly to 8.35 × 10^3^ CFU/g at 20 days. When the weight ratio of pesticide to seed was 1:100, the population of *F. solani* in the rhizosphere soil exhibited a gradual declining trend 10 days post-inoculation, with a more rapid decrease. On the 0th and first days, the *F. solani* counts were 4.09 × 10^4^ CFU/g and 1.02 × 10^4^ CFU/g, respectively. On the third day, it reached its lowest point of 8.11 × 10^3^ CFU/g. By the fifth and tenth days, the *F. solani* population in the rhizosphere soil stabilized at 9.8 × 10^3^ CFU/g and 9.11 × 10^3^ CFU/g, respectively. Additionally, on the 20th day post-inoculation, the *F. solani* count in the rhizosphere soil slightly increased, reaching 1.31 × 10^4^ CFU/g. In the pathogen control, *F. solani* showed a gradual declining trend within 20 days, and the final number was 2.86 × 10^4^ CFU/g (Figure 5a). The results indicated that coating with different proportions of drugs could inhibit *F. solani*, and the population of *F. solani* remained low 20 days after inoculation, showing a significant difference compared to the pathogen control. The treatment with an agent/seed weight ratio of 1:100 reduced the population of *F. solani* in the rhizosphere soil of the cucumber more rapidly, thereby protecting the healthy growth of the cucumber.

The changes in the quantity of *P. polymyxa* ZF129^Rif^ in the cucumber rhizosphere soil and roots within 20 days after inoculation were also measured. When the agent/seed weight ratio was 1:10, the population of *P. polymyxa* ZF129^Rif^ in the rhizosphere soil exhibited a gradual increasing trend 5 days post-inoculation. On the 0th, first, and third days, the *P. polymyxa* ZF129^Rif^ counts were 3.17 × 10^3^ CFU/g, 1.17 × 10^4^ CFU/g, and 1.73 × 10^4^ CFU/g, respectively. On the fifth day, it reached its highest point of 3.66 × 10^4^ CFU/g. By the 10th and 20th days, the *P. polymyxa* ZF129^Rif^ population in the rhizosphere soil decreased to 2.44 × 10^4^ CFU/g and 1.7 × 10^4^ CFU/g, respectively. When the weight ratio of pesticide to seed was 1:100, the number of *P. polymyxa* ZF129^Rif^ in the rhizosphere soil gradually decreased from 3.63 × 10^3^ CFU/g to 4.37 × 10^4^ CFU/g; from 0 to 10 days, the quantity increased more than tenfold. The numbers of *P. polymyxa* ZF129^Rif^ on the first, third, and fifth days were 1.26 × 10^4^ CFU/g, 2.05 × 10^4^ CFU/g, and 2.82 × 10^4^ CFU/g, respectively. Then, the number of *P. polymyxa* ZF129^Rif^ in the rhizosphere soil increased to 3.09 × 10^4^ CFU/g at 20 days (Figure 5b). The results indicated that the variation in the population of *P. polymyxa* ZF129^Rif^ in the cucumber rhizosphere soil exhibited a consistent trend of initial increase, followed by a decrease across the different drug ratios. Notably, the decline in the *P. polymyxa* ZF129^Rif^ population was more gradual in the 1:100-drug ratio treatment, allowing for a prolonged protective effect.

When the agent/seed weight ratio was 1:10 and 1:100, *P. polymyxa* ZF129^Rif^ could not be detected in the cucumber roots three days before inoculation, indicating that it had not yet established itself in the cucumber roots. Starting from the fifth day, *P Polymyxa* ZF129^Rif^ could only be detected when the agent/seed weight ratio was 1:10 and 1:100, where the number of *Polymyxa* ZF129^Rif^ was 9.43 × 10^3^ CFU/g and 4.73 × 10^3^ CFU/g, respectively.

With the increase in time, the number of *Polymyxa* ZF129^Rif^ in the cucumber roots decreased; this may be due to the depletion of nutrients within ZF129 itself or competition with the local microbiota. When the ratio of the agent/seed weight was 1:10, the number of *Polymyxa* ZF129^Rif^ on the tenth and twentieth days was 7.33 × 10^3^ CFU/g and 5.17 × 10^3^ CFU/g, whereas when the ratio of the agent/seed weight was 1:100, the number of *Polymyxa* ZF129^Rif^ on the tenth and twentieth days was 3.96 × 10^3^ CFU/g and 3.33 × 10^3^ CFU/g, which indicated that the inhibitory effect of *P. polymyxa* ZF129^Rif^ on *F. solani* would gradually decline with the increase in time (Figure 5c).

The alterations in the quantity of *P. polymyxa* ZF129^Rif^ and *F. solani* in the cucumber rhizosphere soil post-coating were negatively correlated, confirming that the proliferation of *P. polymyxa* ZF129 can effectively regulate the increase in *F. solani* (Figure 6).

Scanning electron microscopy images of the cucumber root system (Figure 7) allowed us to observe rod-shaped ZF129 in the root system, with a length of approximately 2 μm, and to observe the membranous material, similar to the biofilm formation structure. It was further verified that *P. polymyxa* ZF129 could stably colonize the cucumber root system.

### 3.7. Changes in Rhizosphere Soil Enzyme Activity and Root Defense Enzyme Activity of Cucumbers

Soil enzyme activity is an important indicator of soil ecosystem health and function. In this work, four types of soil enzymes, such as urease (S-UE), sucrose (S-SC), alkaline phosphatase (S-AKP/ALP), and Catalase (S-CAT), were investigated in the abovementioned four treatments (Figure 8).

There were higher S-UE and S-SC activities in the soil infected with *F. solani*. Compared with the *F. solani* control (T4), coating the seeds with ZF129 significantly improved the S-UE activity by 38.82% (T1) and 13.92% (T2). S-UE can catalyze the hydrolysis of urea into ammonia (NH_3_) and carbon dioxide, and the released ammonia is converted into nitrate (NO_3_^–^) through nitrification, providing the main nitrogen source for plants (accounting for 40–60% of their nitrogen requirements). However, ZF129 did not promote an increase in the activity of the other three soil enzymes (S-SC, S-AKP/ALP, and S-CAT). This indicates that ZF129 cannot play a positive role in promoting the secretion of plant growth factors and stimulating microbial activity.

As it was known that many biocontrol bacteria and pathogens can induce a plant defense response to fungal, bacterial, and viral diseases, in the present study, the enzymatic activities of POD, SOD, and CAT in the roots of the cucumbers induced by ZF129 and infected with *F. solani* were examined. The results revealed that roots of T1 and T2 showed higher CAT and POD enzyme activities than in T3 and T4. No significant difference in the SOD enzyme activity was detected across all of the treatments in the comparison (Figure 9).

## 4. Discussion

Microbial seed coating represents an environmentally friendly method for controlling soil-borne diseases [27,28,29]. How to enable microorganisms to colonize in the rhizosphere and better exert their biocontrol effects is currently an important challenge in this field [30,31,32]. In this study, we provided an effective strategy for the seed-coating application of *P. polymyxa* ZF129 by microencapsulation in spray-dried gum arabic. The obtained oil-based seed coating agent-SC ZF129m showed a much better storage stability than in the control group (ZF129 cell culture), indicating that even with the many adjuvants in the formulation, the arabic gum can still improve the survival rate of the ZF129 cells. Although previous studies have reported the protective effect of arabic gum on biocontrol bacteria, Srivastava et al. (2010) used talcum powder and arabic gum to prepare a fluorescent pseudomonas seed coating agent to coat tomato seeds, which played an initiating role in the tomato seeds, significantly reduced the time required for seed germination, and reduced the incidence rates of *fusarium* wilt in the pot and field experiments [33]. However, they were all applied as dry products rather than oil suspension formulations.

The stable colonization of biocontrol strains on plants is crucial for their biocontrol effects [34]. There are many factors that affect the success of microbial colonization, such as the ability to form biofilms and competitiveness with pathogens [35,36,37]. In our pot experiments, the results showed that the ZF129 microcapsules not only inhibited the proliferation of pathogenic bacteria in the rhizosphere soil, but also promoted the proliferation of biocontrol bacteria in the soil. At the same time, we also observed the colonization of biocontrol bacteria in the roots of the cucumber seedlings, which further confirmed our above conclusion. In addition, we speculate that arabic gum may not only protect the ZF129 cells, but also promote their colonization in the rhizosphere, which requires further study in the future. Similarly, in a previous study, arabic gum has been described as a protector of *Trichoderma* conidia by microencapsulation, increasing the survival of *T. asperellum* and *T. harzianum* subjected to temperatures up to 90 °C. Furthermore, the treatment of chickpea seeds with gum arabic and *T. harzianum* conidia was found to increase *Trichoderma* root colonization and to improve plant survival [38].

In our pot experiment, the ZF129m-seeds not only showed the effectiveness of the control of cucumber root rot, but also enhanced the growth of the cucumber seedlings. It was acknowledged that *P. polymyxa* can produce a variety of defensive enzymes and antibiotics, which can degrade the cell walls of pathogenic microorganisms and affect other cellular structures, thus reducing the disease index [39]. A previous study showed that *P. polymyxa* HX-140 was capable of producing proteases, cellulases, β-1,3-glucanases, and antifungal volatile organic compounds, and that the use of *P. polymyxa* HX-140 bacterial suspensions for rooting resulted in a 55.6% reduction in *Fusarium* wilt in cucumber seedlings in a pot experiment [40]. However, the main antifungal metabolites of ZF129 still need further identification. *P. polymyxa* can also develop a systemic resistance to specific pathogens by inducing the plant activation of defense mechanisms. It was proved that ZF129 induced the plant defense response by improving the CAT and POD enzyme activities. Similarly, *P. polymyxa* SF05 reduced the disease index of Maize Sheath Blight by 37.06% by inducing the expression of resistance genes such as *ZmPR1a*, *OPR1*, and *OPR7* in the maize leaves and stems [41].

As for the growth-promoting effect of ZF129, we speculate that it is attributed to its abilities in phosphate solubilizing and phytohormone production. A functional analysis of *P. polymyxa* ZF129’s genome showed that it has potential genes related to phosphate solubilizing (e.g., pstC, pstA, and pstB) and IAA production (*trp genes*) [19,42]. However, more solid evidence needs to be provided in future studies. On the other hand, inoculation with ZF129 improved the soil urease activity, which also enhanced the growth of the cucumber seedlings. The level of soil enzyme activity can effectively reflect the intensity and direction of various biochemical reactions in the soil, represents the exuberant degree of material metabolism in the soil, and is an important index of the soil fertility. Typically, urease is the most active hydrolase in soil. It is closely related to the number of microorganisms and the content of organic matter in the soil, and can hydrolyze the urea in the soil and release ammonium for crop utilization. Notably, the ZF129 coating did not improve the other soil enzymes’ activities (sucrose, alkaline phosphatase, and catalase). This may be because we only measured these data on the 30th day after sowing, and did not conduct continuous monitoring during this period. Therefore, in the future, further time-dependent experiments are needed to study the effects of the biocontrol bacteria on soil enzyme activity. In addition, considering that the bioassay experiments were performed in greenhouses, there are still some limitations to the study, such as the impact of environmental factors and consistency across plant species.

## 5. Conclusions

In this study, we successfully prepared an oil-based microbial seed treatment agent, in which *P. polymyxa* ZF129 was encapsulated in gum arabic microcapsules and corn oil was used as a disperse medium. The process parameters of preparing the biocontrol bacteria microcapsules by the spray-drying method were optimized. The obtained CS-ZF129 showed good storage stability in term of the cell viability at room temperature. *P. polymyxa* ZF129 was successfully colonized in the cucumber root tissues and rhizosphere soil, which confirmed that *P. polymyxa* ZF129 could act as a sustained antagonist against the pathogenic fungus around the cucumber rhizosphere. Moreover, the CS-ZF129 not only showed effectiveness in the control of cucumber root rot, but also had a growth-promoting effect on the cucumber plants. Future research will continue to focus on improving the storage stability of biocontrol bacteria in different environments and expanding the applications of this formulation in further plant disease control.

## Figures and Tables

**Figure 1 biology-14-00375-f001:**
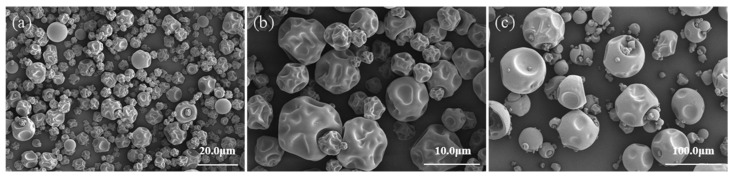
Scanning electron microscope images of *P. polymyxa* ZF129 microcapsules at inlet air temperature of 120 °C. (**a**–**c**) SEM images of *P. polymyxa* ZF129 microcapsules at different magnifications.

**Figure 2 biology-14-00375-f002:**
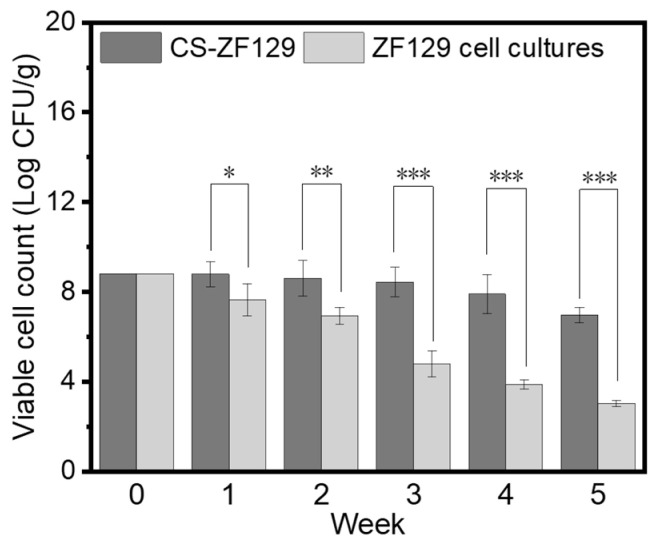
The changes in the biocontrol bacterial contents of the CS-ZF129 and ZF129 cell cultures over time at 25 °C. The error bars in the figure indicate the standard deviations of the three replicate experiments performed for each treatment (*p* < 0.05). Bars labeled with asterisks denote statistical significance compared to the control group (*: *p* < 0.05, **: *p* < 0.01, ***: *p* < 0.001).

**Figure 3 biology-14-00375-f003:**
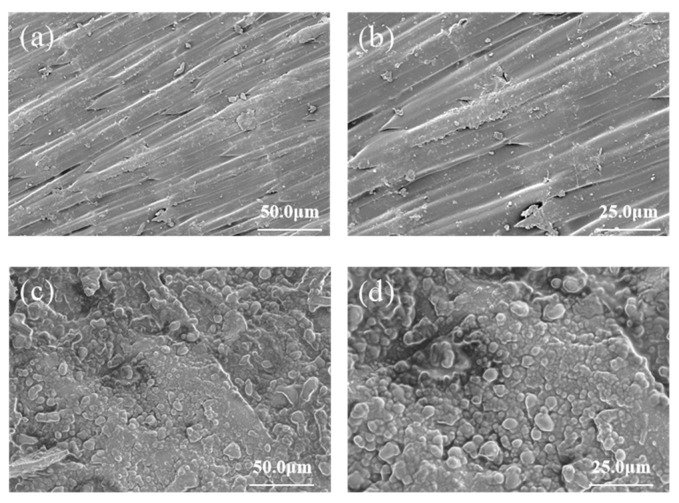
Scanning electron microscope images of cucumber seeds before and after coating. (**a**,**b**) Cucumber seeds before coating at different magnifications; (**c**,**d**) cucumber seeds after coating at different magnifications.

**Figure 4 biology-14-00375-f004:**
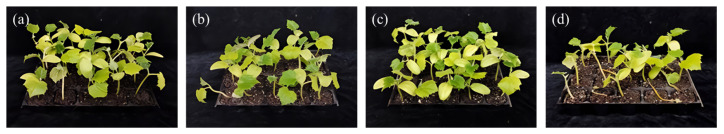
Efficacy of CS-ZF129 against cucumber *Fusarium* root rot: (**a**) T1, (**b**) T2, (**c**) T3, (**d**) T4.

**Figure 5 biology-14-00375-f005:**
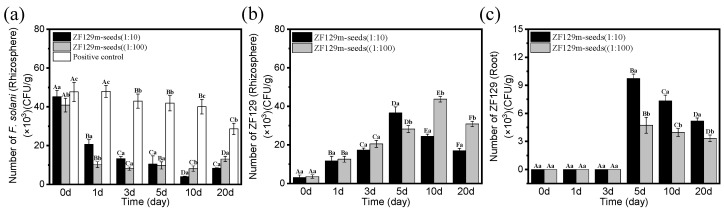
The changes in the numbers of *P. polymyxa* ZF129^Rif^ and *F. solani* in the rhizospheres of the cucumbers after coating with the microcapsule suspensions at an agent/seed weight ratio of 1:10/1:100. (**a**) The changes in the number of *F. solani* in the cucumber rhizosphere soil; (**b**) the changes in the number of ZF129 in the cucumber rhizosphere soil; (**c**) the changes in the number of ZF129 in the cucumber roots. The error bars in the figure indicate the standard deviations of the three repetitions of the experiment, and the different uppercase letters indicate significant differences between the same treatment at different times, while the different lowercase letters indicate significant differences between different treatments on the same day (*p* < 0.05).

**Figure 6 biology-14-00375-f006:**
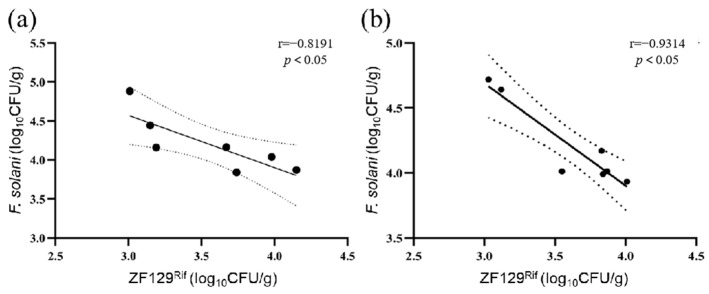
The correlation between the bacterial strain *P. polymyxa* ZF129^Rif^ and *F. solani* in the cucumber rhizosphere soil. (**a**) T1, (**b**) T2.

**Figure 7 biology-14-00375-f007:**
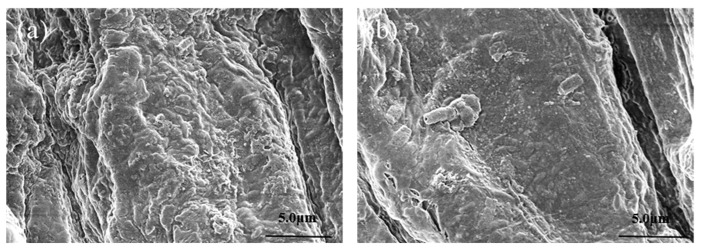
Scanning electron microscope images of *P. polymyxa* ZF129 colonization in cucumber roots. (**a**) T1, (**b**) T2.

**Figure 8 biology-14-00375-f008:**
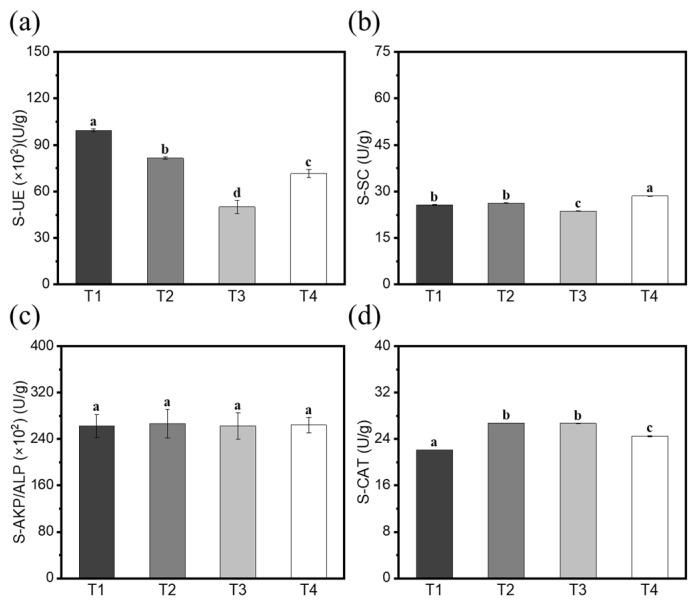
The effects of the CS-ZF129 on the rhizosphere soil enzyme activities of the cucumbers. (**a**) S-UE, (**b**) S-SC, (**c**) S-AKP/ALP, (**d**) S-CAT. The error bars in the figure indicate the standard deviations of the three repetitions of the experiment, and the different letters indicate significant differences in the soil enzyme activities (*p* < 0.05).

**Figure 9 biology-14-00375-f009:**
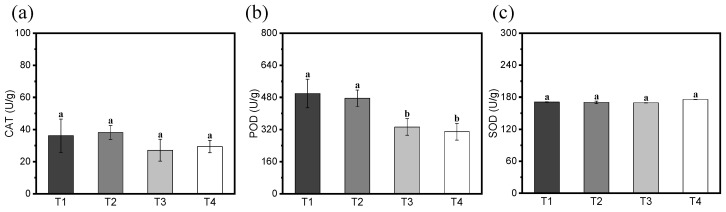
Effects of CS-ZF129 on defense enzyme activities in cucumber roots. (**a**) CAT, (**b**) POD, (**c**) SOD. Error bars in figure indicate standard deviations of three replicates of experiment, and different letters indicate significant differences in defense enzyme activities (*p* < 0.05).

**Table 1 biology-14-00375-t001:** Growth parameters of coated and bare cucumber seeds in pot experiments.

Group	Treatment	Leaf Area (cm^2^)	Chlorophyll (SPDA)	Shoot Height (cm)	Root Length (cm)	Shoot Fresh Weight (g)	Root Fresh Weight (g)
T1	ZF129m-seeds (coating agent/seed ratio = 1:10)	44.57 ± 7.15 ^a^	39.33 ± 2.07 ^b^	14.52 ± 2.44 ^ab^	12.38 ± 0.10 ^a^	24.93 ± 0.23 ^b^	16.39 ± 0.61 ^a^
T2	ZF129m-seeds (coating agent/seed ratio = 1:100)	46.22 ± 2.64 ^a^	38.37 ± 1.30 ^b^	11.90 ± 0.88 ^b^	12.75 ± 0.64 ^a^	24.35 ± 0.65 ^b^	9.48 ± 0.21 ^d^
CK	Bare seeds (Non-treated control)	41.00 ± 7.06 ^a^	38.60 ± 1.92 ^b^	11.82 ± 0.84 ^b^	8.90 ± 1.08 ^b^	19.67 ± 0.28 ^d^	8.66 ± 0.56 ^d^

Note: Different lowercase letters represent differences in leaf area, plant height, root length and other aspects among different treatments (*p* < 0.05).

**Table 2 biology-14-00375-t002:** The emergence rates of ZF129-coated seeds with different ratios and their control effects on cucumber *Fusarium* root rot.

Group	Treatment	Emergence Rate (%)	Disease Index	Control Efficacy (%)
T1	ZF129m-seeds (coating agent/seed ratio = 1:10)	89.63 ± 1.35 ^a^	49.98 ± 6.30 ^a^	29.01 ± 0.09 ^a^
T2	ZF129m-seeds (coating agent/seed ratio = 1:100)	91.42 ± 1.12 ^a^	36.30 ± 1.25 ^b^	46.30 ± 0.02 ^b^
T3	Bare seeds (Non-treated control)	93.55 ± 2.84 ^a^	—	—
T4	Bare seeds (Pathogen control)	83.55 ± 1.64 ^b^	68.00 ± 1.91 ^c^	—

Note: Different lowercase letters represent differences in emergence rate, disease index and other aspects among different treatments (*p* < 0.05).

## Data Availability

The original contributions presented in this study are included in the article. Further inquiries can be directed to the corresponding author(s).

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
