# Peer review of "Coating Seeds with Paenibacillus polymyxa ZF129 Microcapsule Suspension Enhanced Control Effect on Fusarium Root Rot and Promoted Seedling Growth in Cucumber"

_biology, 2025, doi:10.3390/biology14040375_

Round 1
Reviewer 1 Report
Comments and Suggestions for Authors
This manuscript presents a method with significant practical and environmental value for controlling Fusarium root rot through cucumber seed coating. However, there are a few concerns need to be addressed.
Major Concerns:
- This manuscript requires revisions to improve language fluency and logical coherence.
- Most importantly, the article does not clearly and explicitly highlight its innovation. What is the main novelty that this manuscript aims to focus on among the three fundamental components: microorganisms, carriers, and additional additives? Is it P. polymyxa ZF129, the microcapsule suspension, the spray drying method, or a combination of these?
- How can you determine whether the enhancement is due to coating with P. polymyxa ZF129 microcapsule suspension or P. polymyxa ZF129 itself? Have you compared it with other carriers containing P. polymyxa ZF129?
- The current wording makes it unclear whether the novelty and effect of CS-ZF129 come from ZF129 itself or whether the CS method is responsible for disease control while maintaining ZF129’s effectiveness.
Simple Summary:
- The information is too fragmented, lacks fluency, and does not effectively highlight the key points of the study.
Abstract:
- This section has the same issues as the "Simple Summary" in which the information is too fragmented, lacks fluency, and does not effectively highlight the key points of the study, and improvement is needed.
- Lines 26-29: Experiments are mentioned in this part, but there aren’t any conclusions drawn from the results mentioned.
Introduction:
- Lines 44-62: Insufficient background information on previous studies regarding the application of microcapsule suspension in seed coating.
- Lines 63-80: More details are needed on previous research related to P. polymyxa ZF129.
- Lines 81-91: These lines suffer from the same issues as the "Simple Summary" in which the information is too fragmented, lacks fluency, and does not effectively highlight the key points of the study, and improvement is needed.
Materials and Methods:
- Line 125: What does "different carrier" refer to in this manuscript? Please clarify.
Results:
- Figure 2: The figure lacks significance markers.
- Lines 252-254: The sentence "CS-ZF129 can significantly improve P. polymyxa ZF129 has the ability to survive" contains a grammatical error, making the meaning unclear and fragmented.
- Figure 3, Lines 267-268: What do a and b refer to, respectively? Similarly, what do c and d refer to?
- Lines 280-281: The statement "The growth promotion effect of ZF129 was attributed to its secondary metabolites such as IAA" needs supporting evidence. On what basis is this inference made? Moreover, any results not presented in the Results section should not be included here and should be placed in the Discussion section instead.
- Table 2: No significance markers are provided for the “Disease Index”.
- Lines 293-294: The statement "This may be because the high rate of coating affected germination" belongs in the Discussion section rather than the Results section. Furthermore, did you measure the germination rate with and without coating? If not, how was this assumption made?
Discussion:
- Lines 400-401: Why reference paper 28 in a sentence summarizing the findings of this study?
- Lines 404-406: Please specify which studies the phrase "previous studies" refers to.
This manuscript requires revisions to improve language fluency and logical coherence.
Author Response
Referee: 1
This manuscript presents a method with significant practical and environmental value for controlling Fusarium root rot through cucumber seed coating. However, there are a few concerns need to be addressed.
Response: We sincerely thank the reviewer for the positive comments. We have revised the manuscript according to the suggestions of Reviewers. We hope that the revised manuscript will be published finally and make our contribution in the research field of control of cucumber Fusarium root rot.
Comments 1: Most importantly, the article does not clearly and explicitly highlight its innovation. What is the main novelty that this manuscript aims to focus on among the three fundamental components: microorganisms, carriers, and additional additives? Is it P. polymyxa ZF129, the microcapsule suspension, the spray drying method, or a combination of these?
Response 1: We thank the reviewer for the helpful comments. The main novelty of this manuscript is the formulation of P. polymyxa ZF129. The formulation process incorporation two critical technological innovations: (1) Precise encapsulation of ZF129 spores through spray-drying technology using gum arabic as the protective matrix, and (2) Strategic suspension of the obtained microcapsules in corn oil. We highlighted the novelty in the line113-116:
Line 113-116:
The formulation process incorporated two critical technological innovations: (1) Pre-cise encapsulation of ZF129 spores through spray-drying technology using gum arabic as the protective matrix, and (2) Strategic suspension of the obtained microcapsules in corn oil supplemented.
Comments 2: How can you determine whether the enhancement is due to coating with P. polymyxa ZF129 microcapsule suspension or P. polymyxa ZF129 itself? Have you compared it with other carriers containing P. polymyxa ZF129?
Response 2: Thank the reviewer for the comment. We investigated the storage stability of the CS-ZF129 microcapsule and ZF129 cell cultures. The initial microbial content was 6×108 CFU/g, when stored at 25°C, the biocontrol bacterial content was 3.76×108 CFU/mL after 2 weeks of storage, which was 50% lower than the initial biocontrol bacterial content. The result indicates that microencapsulation can significantly improve the survival ability of P. Polymyxa ZF129. The data was shown in Section 3.2.
We believe there other carrier can also protect the biocontrol bacterial. However, we did not compare the gum Arabic with other carriers, because many factors need to be considered, not only about their protective effects but also related with the method of microencapsulation and their costs.
Comments 3: The current wording makes it unclear whether the novelty and effect of CS-ZF129 come from ZF129 itself or whether the CS method is responsible for disease control while maintaining ZF129’s effectiveness.
Response 3: Thank you for pointing this out. In our previous research, ZF129 itself had a certain preventive effect on cucumber root rot disease, and after being prepared into CS-ZF129, its customization ability in cucumber roots was enhanced, thereby further improving the control effect on cucumber root rot disease.
Comments 4: The information is too fragmented, lacks fluency, and does not effectively highlight the key points of the study.
Response 4: We thank the reviewer for the comments. We have reorganized the simple summary to effectively highlight the key points of the research (Lines 10-18).
Line 10-18:
In this study, a Paenibacillus polymyxa ZF129 microcapsule suspension (CS-ZF129) was prepared. P. polymyxa ZF129 was microencapsulated by gum arabic via the spray-drying method. Coating seeds with CS-ZF129 not only increased the control efficacy of cucumber root rot to 46.30 ± 0.02%, but also promoted the growth of cu-cumber roots and other aspects, effectively improving the customization ability of P. polymyxa ZF129 in cucumber root system was analyzed to clarify the biological control mechanism of cucumber Fusarium root rot disease through enzyme activity. This study provides an efficient means based on seed coating for the biological control of cucumber Fusarium root rot.
Comments 5: This section has the same issues as the "Simple Summary" in which the information is too fragmented, lacks fluency, and does not effectively highlight the key points of the study, and improvement is needed.
Response 5: Thank you for pointing this out. We agree with this comment. Therefore, we have reorganized the abstract to effectively highlight the key points of the research (Lines 19-31).
Comments 6: Lines 26-29: Experiments are mentioned in this part, but there aren’t any conclusions drawn from the results mentioned.
Response 6: Thank you for pointing this out. We have supplemented the experimental conclusion (Lines 22-25).
Comments 7: Lines 44-62: Insufficient background information on previous studies regarding the application of microcapsule suspension in seed coating.
Response 7: We thank the reviewer for the helpful comments. We have supplemented the background information of the preliminary research on the application of microcapsule suspension in seed coating (Lines 80-87).
Comments 8: Lines 63-80: More details are needed on previous research related to P. polymyxa ZF129.
Response 8: Thank you for pointing this out. We have supplemented the detailed information of previous studies related to P. polymyxa ZF129 (Lines 104-110).
Comments 9: Lines 81-91: These lines suffer from the same issues as the "Simple Summary" in which the information is too fragmented, lacks fluency, and does not effectively highlight the key points of the study, and improvement is needed.
Response 9: We thank the reviewer for the helpful comments. We have rewritten this section to ensure its fluency and effectively highlight the key points of the research (Lines 111-121).
Comments 10: Line 125: What does "different carrier" refer to in this manuscript? Please clarify.
Response 10: Thank you for pointing this out. We have provided additional explanations for different carriers (Lines 156).
Comments 11: Figure 2: The figure lacks significance markers.
Response 11: Thank you for pointing this out. We have made modifications to the image and made significance markers.
Comments 12: Lines 252-254: The sentence "CS-ZF129 can significantly improve P. polymyxa ZF129 has the ability to survive" contains a grammatical error, making the meaning unclear and fragmented.
Response 12: Thank you for pointing this out. We have made modifications to the sentence (Lines 295-296).
Comments 13: Figure 3, Lines 267-268: What do a and b refer to, respectively? Similarly, what do c and d refer to?
Response 13: Thank you for pointing this out. We have already explained in the caption that a and b are bare cucumber seeds, and c and d are ZF129m-seed.
Comments 14: Lines 280-281: The statement "The growth promotion effect of ZF129 was attributed to its secondary metabolites such as IAA" needs supporting evidence. On what basis is this inference made? Moreover, any results not presented in the Results section should not be included here and should be placed in the Discussion section instead.
Response 14: Thank you for pointing this out. We have removed the unclassified item and moved it to the discussion section.
Comments 15: Table 2: No significance markers are provided for the “Disease Index”.
Response 15: Thank you for pointing this out. We have supplemented the significance of the disease index in Table 2.
Comments 16: Lines 293-294: The statement "This may be because the high rate of coating affected germination" belongs in the Discussion section rather than the Results section. Furthermore, did you measure the germination rate with and without coating? If not, how was this assumption made?
Response 16: Thank you for pointing this out. We measured the germination rate of coated and uncoated cucumber seeds and found that coating had a slight impact on germination rate. The relevant data has been supplemented in Table 2.
Comments 17: Lines 400-401: Why reference paper 28 in a sentence summarizing the findings of this study?
Response 17: Thank you for pointing this out. We found an error in the citation position of the literature through proofreading, and now we have made modifications to the citation position of the literature.
Comments 18: Lines 404-406: Please specify which studies the phrase "previous studies" refers to.
Response 18: Thank you for pointing this out. We have supplemented the research in "previous studies" (Lines 468-471).

Reviewer 2 Report
Comments and Suggestions for Authors
Dear Authors,
Your work is promising. It is however lacking some key details and the wording is sometimes confusing. I have taken time to comment on most lines and would suggest you improve on them and ensure the reader gets the most of this paper should you consider to resubmit it or repeat the studies
Key issues include:
Abstract
-
-
Lines 18-29:
-
The abstract lacks specific quantitative results (e.g., percentage reduction in disease severity, growth promotion metrics).
-
The biocontrol mechanism is mentioned but not explained in detail.
Presentation: Figures and tables require clearer labeling and annotations to improve readability.
-
-
-
Introduction
-
Lines 34-43:
-
The introduction does not clearly state the novelty of the study compared to existing literature on biocontrol agents.
-
The rationale for using P. polymyxa ZF129 specifically is not well-justified.
-
-
Lines 44-55:
-
The discussion of chemical pesticides and their limitations is too general and lacks specific references to recent studies.
-
-
Lines 63-80:
-
The background on P. polymyxa is extensive but somewhat repetitive. Consider condensing this section to focus on the most relevant studies.
-
Materials and Methods
-
Lines 93-100:
-
The list of materials is overly detailed and includes unnecessary information (e.g., purity of sodium chloride). Focus on key reagents and their relevance to the study.
-
-
Lines 117-129:
-
The spray-drying method is described, but critical parameters (e.g., nozzle size, airflow rate) are missing, which are essential for reproducibility.
-
-
Lines 130-135:
-
The FESEM procedure lacks details on sample preparation (e.g., drying conditions, gold coating thickness).
-
-
Lines 136-146:
-
The seed coating procedure is described, but the rationale for using specific adjuvants (e.g., Tween 80, blue coloring agent) is not explained.
-
-
Lines 147-157:
-
The storage stability experiment lacks details on the storage conditions (e.g., humidity, light exposure).
-
-
Lines 158-167:
-
The growth promotion experiment does not specify the soil type or nutrient content, which could influence the results.
-
-
Lines 168-181:
-
The disease severity index (DSI) calculation is described, but the scoring criteria are not clearly defined (e.g., how "localized wilting" is distinguished from "overall wilting").
-
-
Lines 182-201:
-
The rifampicin resistance induction process is described, but the potential impact of rifampicin resistance on ZF129's biocontrol efficacy is not discussed.
-
-
Lines 202-210:
-
The real-time PCR protocol references a previous study but does not provide sufficient details (e.g., primer sequences, reaction conditions).
-
-
Lines 211-220:
-
The enzyme activity assays are referenced to a kit, but no specific details (e.g., incubation time, temperature) are provided.
-
Results
-
Lines 227-236:
-
The SEM images (Figure 1) lack scale bars, making it difficult to interpret the size of the microcapsules.
-
-
Lines 241-254:
-
The storage stability data (Figure 2) is presented without statistical analysis (e.g., significance of differences between time points).
-
-
Lines 259-265:
-
The SEM images of coated seeds (Figure 3) are not quantitatively analyzed (e.g., coating thickness, uniformity).
-
-
Lines 269-281:
-
The growth promotion data (Table 1) lacks units for some parameters (e.g., chlorophyll content, shoot weight).
-
-
Lines 283-295:
-
The disease control efficacy data (Table 2) is not statistically compared between treatments (e.g., 1:10 vs. 1:100 coating ratios).
-
-
Lines 299-324:
-
The correlation between ZF129Rif and F. solani populations is mentioned, but no statistical test (e.g., Pearson correlation) is reported.
-
-
Lines 325-341:
-
The colonization data (Figure 5) is not discussed in the context of other studies on microbial colonization in plant roots.
-
-
Lines 342-353:
-
The decline in ZF129Rif populations over time is not explained mechanistically (e.g., competition with native microbiota, nutrient depletion).
-
-
Lines 354-360:
-
The SEM images of ZF129 colonization in roots (Figure 7) are not quantitatively analyzed (e.g., number of cells per unit area).
-
-
Lines 372-385:
-
The enzyme activity data (Figure 8) is presented without discussion of its biological significance (e.g., how urease activity contributes to plant growth).
-
-
Lines 386-395:
-
The defense enzyme data (Figure 9) is not compared to other studies on induced systemic resistance (ISR) in plants.
-
Discussion
-
Lines 396-415:
-
The discussion does not adequately address the limitations of the study (e.g., impact of environmental factors, consistency across plant species).
-
-
Lines 416-430:
-
The discussion of ZF129's biocontrol mechanisms is speculative and lacks experimental evidence (e.g., identification of specific antifungal metabolites).
-
-
Lines 431-446:
-
The discussion of ZF129's growth-promoting effects is not supported by mechanistic data (e.g., phosphate solubilization, hormone production).
-
Conclusions
-
Lines 447-460:
-
The conclusions are overly broad and not fully supported by the data (e.g., claims about industrial scalability without evidence).
-
General Issues
-
Figures and Tables:
-
Many figures and tables lack clear labels, annotations, or units (e.g., Figure 2, Table 1).
-
Error bars or standard deviations are missing in some figures (e.g., Figure 5).
-
-
Terminology:
-
Key terms (e.g., "colonization ability," "control efficacy") are used inconsistently or without clear definitions.
-
-
Ethical Considerations:
-
The use of rifampicin-resistant strains is not discussed in the context of potential risks (e.g., antibiotic resistance gene transfer).
-
-

Whereas I could easily read and follow the paper, It needs to be coherent.
Author Response
Referee: 2
Your work is promising. It is however lacking some key details and the wording is sometimes confusing. I have taken time to comment on most lines and would suggest you improve on them and ensure the reader gets the most of this paper should you consider to resubmit it or repeat the studies
Response: We express our profound gratitude to the reviewers for their constructive evaluation and endorsement of this work's publication merits. In response to the valuable feedback, we have meticulously incorporated all suggested improvements into the revised manuscript, particularly enhancing the methodological details of microbial encapsulation and expanding the discussion on rhizosphere colonization dynamics. We trust these refinements will advance the scientific understanding of biological control strategies against Fusarium root rot in cucurbit crops. It is our sincere hope that this enhanced version meets the journal's rigorous standards for dissemination, thereby contributing substantively to sustainable plant disease management research.
Comments 1: Lines 18-29: The abstract lacks specific quantitative results (e.g., percentage reduction in disease severity, growth promotion metrics).
Response 1: Thank you for pointing this out. We have supplemented the data description of disease prevention and control, as well as the effect of promoting growth (Lines 22-25).
Comments 2: Lines 18-29: The biocontrol mechanism is mentioned but not explained in detail.
Response 2: Thank you for pointing this out. The biological control mechanism was explained in subsequent discussions, such as the production of multiple enzymes by Bacillus subtilis (Lines 491-495).
Comments 3: Lines 18-29: Presentation: Figures and tables require clearer labeling and annotations to improve readability.
Response 3: Thank you for pointing this out. We have made modifications to the labels and annotations of the figures and tables to improve their readability.
Comments 4: Lines 34-43: The introduction does not clearly state the novelty of the study compared to existing literature on biocontrol agents.
Response 4: Thank you for pointing this out. We will discuss the novelty and necessity of our research compared to existing biological control methods in the future (Lines 67-74).
Comments 5: Lines 34-43: The rationale for using P. polymyxa ZF129 specifically is not well-justified.
Response 5: Thank you for pointing this out. We have supplemented the importance of using P. polymyxa ZF129 (Lines 104-109).
Comments 6: Lines 44-55: The discussion of chemical pesticides and their limitations is too general and lacks specific references to recent studies.
Response 6: Thank you for pointing this out. I have re discussed chemical insecticides and supplemented recent research (Lines 60-64).
Comments 7: Lines 63-80: The background on P. polymyxa is extensive but somewhat repetitive. Consider condensing this section to focus on the most relevant studies.
Response 7: Thank you for pointing this out. We have rewritten the high section and streamlined the reference section (Lines 88-110).
Comments 8: Lines 93-100: The list of materials is overly detailed and includes unnecessary information (e.g., purity of sodium chloride). Focus on key reagents and their relevance to the study.
Response 8: Thank you for pointing this out. We have made modifications to the material list and removed unnecessary information (Lines 128-130).
Comments 9: Lines 117-129: The spray-drying method is described, but critical parameters (e.g., nozzle size, airflow rate) are missing, which are essential for reproducibility.
Response 9: Thank you for pointing this out. We have supplemented the parameters for preparing P. polymyxa ZF129 microcapsule powder by spray drying method (Lines 150-153).
Comments 10: Lines 130-135: The FESEM procedure lacks details on sample preparation (e.g., drying conditions, gold coating thickness).
Response 10: Thank you for pointing this out. We have supplemented the details of FESEM program in sample preparation (Lines 161-168).
Comments 11: Lines 136-146: The seed coating procedure is described, but the rationale for using specific adjuvants (e.g., Tween 80, blue coloring agent) is not explained.
Response 11: Thank you for pointing this out. The c specific adjuvants (e.g., Tween 80, blue coloring agent) used in the seed coating process are the result of our previous skim screening, and the specific screening process is quite complex and has not been demonstrated.
Comments 12: Lines 147-157: The storage stability experiment lacks details on the storage conditions (e.g., humidity, light exposure).
Response 12: Thank you for pointing this out. We have supplemented the storage stability experiment with detailed information on storage conditions (Lines 181-184).
Comments 13: Lines 158-167: The growth promotion experiment does not specify the soil type or nutrient content, which could influence the results.
Response 13: Thank you for pointing this out. We have provided an explanation of the soil types used in the growth promotion experiment (Lines 194-195).
Comments 14: Lines 168-181: The disease severity index (DSI) calculation is described, but the scoring criteria are not clearly defined (e.g., how "localized wilting" is distinguished from "overall wilting").
Response 14: Thank you for pointing this out. We have provided additional clarification on the evaluation criteria for the Disease Severity Index (Lines 211-218).
Comments 15: Lines 182-201: The rifampicin resistance induction process is described, but the potential impact of rifampicin resistance on ZF129's biocontrol efficacy is not discussed.
Response 15: Thank you for pointing this out. We also conducted corresponding plate confrontation experiments and field control experiments on ZF129 before and after domestication with rifampicin, and found no difference between the two. Therefore, we cannot provide a detailed description here.
Comments 16: Lines 202-210: The real-time PCR protocol references a previous study but does not provide sufficient details (e.g., primer sequences, reaction conditions).
Response 16: Thank you for pointing this out. We have provided sufficient detailed explanations for the PCR protocol (e.g., primer sequences, reaction conditions) (Lines 239-248).
Comments 17: Lines 211-220: The enzyme activity assays are referenced to a kit, but no specific details (e.g., incubation time, temperature) are provided.
Response 17: Thank you for pointing this out. We have supplemented the specific details of enzyme activity detection (e.g., incubation time, temperature) (Lines 263).
Comments 18: Lines 227-236: The SEM images (Figure 1) lack scale bars, making it difficult to interpret the size of the microcapsules.
Response 18: Thank you for pointing this out. The SEM image (Figure 1) has a scale at the bottom of the image.
Comments 19: Lines 241-254: The storage stability data (Figure 2) is presented without statistical analysis (e.g., significance of differences between time points).
Response 19: Thank you for pointing this out. We conducted statistical analysis on the storage stability data (Figure 2) to supplement the analysis (e.g., significance of differences between time points).
Comments 20: Lines 259-265: The SEM images of coated seeds (Figure 3) are not quantitatively analyzed (e.g., coating thickness, uniformity).
Response 20: Thank you for pointing this out. We conducted quantitative analysis on the coating seeds (Figure 3) in SEM images (e.g., Pearson correlation) (Lines 307).
Comments 21: Lines 269-281: The growth promotion data (Table 1) lacks units for some parameters (e.g., chlorophyll content, shoot weight).
Response 21: Thank you for pointing this out. We have supplemented the parameters units in the growth promotion data (Table 1) (e.g., chlorophyll content, shoot weight). Comparison of SPDA as a relative value of chlorophyll between different treatments.
Comments 22: Lines 283-295: The disease control efficacy data (Table 2) is not statistically compared between treatments (e.g., 1:10 vs. 1:100 coating ratios).
Response 22: Thank you for pointing this out. Our disease control effectiveness data (Table 2) will be supplemented with statistical analysis between treatments.
Comments 23: Lines 299-324: The correlation between ZF129Rif and F. solani populations is mentioned, but no statistical test (e.g., Pearson correlation) is reported.
Response 23: Thank you for pointing this out. The alterations in the quantity of P. polymyxa ZF129Rif and F. solani in the cucumber rhizosphere soil post coating were negatively correlated (Figure 6), report statistical testing is also indicated in the figure.
Comments 24: Lines 325-341: The colonization data (Figure 5) is not discussed in the context of other studies on microbial colonization in plant roots.
Response 24: We thank the reviewer for the comments. More discussion was added in line 482-486.
line 482-486:
Similarly, in a previous study, arabic gum has been described as a protector of Trichoderma conidia by microencapsulation, increasing the survival of T. asperellum and T. harzianum subjected to temperatures up to 90 °C. Besides, the treatment of chickpea seeds with gum arabic and T. harzianum conidia was found to increase Trichoderma-root colonization and to improve plant survival [38]
Comments 25: Lines 342-353: The decline in ZF129Rif populations over time is not explained mechanistically (e.g., competition with native microbiota, nutrient depletion).
Response 25: Thank you for pointing this out. We have provided additional explanations for the mechanism of population decline in ZF129Rif over time (Lines 395-397).
Comments 26: Lines 354-360: The SEM images of ZF129 colonization in roots (Figure 7) are not quantitatively analyzed (e.g., number of cells per unit area).
Response 26: Thank you for pointing this out. We discussed and supplemented the biological significance of enzyme activity data (Lines 426-439).
Comments 27: Lines 372-385: The enzyme activity data (Figure 8) is presented without discussion of its biological significance (e.g., how urease activity contributes to plant growth).
Response 27: Thank you for pointing this out. Due to the uneven colonization of ZF129 in cucumber roots and the significant differences in the number of P. polymyxa species in each field, it cannot be used as a reference. Therefore, a detailed analysis will not be conducted here.
Comments 28: Lines 386-395: The defense enzyme data (Figure 9) is not compared to other studies on induced systemic resistance (ISR) in plants.
Response 28: Thank you for pointing this out. Here we compare the differences between different treatments of defense enzymes.
Comments 29: Lines 396-415: The discussion does not adequately address the limitations of the study (e.g., impact of environmental factors, consistency across plant species).
Response 29: Thank you for the comment. The limitations of the study were discussed as suggested:
Line 517-520:
Besides, considering that the bioassay experiments were performed in greenhouses, there are still some limitations in the study, such as the impact of environmental factors and consistency across plant species.
Comments 30: Lines 416-430: The discussion of ZF129's biocontrol mechanisms is speculative and lacks experimental evidence (e.g., identification of specific antifungal metabolites).
Response 30: Thank you for the comments. In the manuscript, we have mentioned that the main antifungal metabolites of ZF129 still need further identification.(line 426-439). These works will be done in the future.
Comments 31: Lines 431-446: The discussion of ZF129's growth-promoting effects is not supported by mechanistic data (e.g., phosphate solubilization, hormone production).
Response 31: Thank you for the comments. We don’t have very solid data now but we provided the references (41, 42).
Comments 32: Lines 447-460: The conclusions are overly broad and not fully supported by the data (e.g., claims about industrial scalability without evidence).
Response 32: Thank you for the comments. The conclusions was revised as follows:
Line 522-533
“In this study, we successfully prepared an oil-based microbial seed treatment agent, in which P. polymyxa ZF129 was encapsulated in gum arabic microcapsule and corn oil was used as disperse medium. The process parameters of preparing biocontrol bacteria microcapsules by spray drying method were optimized. The obtained CS-ZF129 showed good storage stability in term of cell viability at room temperature. P. polymyxa ZF129 successfully colonized in cucumber root tissues and rhizosphere soil, which confirmed that P. polymyxa ZF129 could act as a sustained antagonist against the pathogenic fungus around the cucumber rhizosphere. Moreover, CS-ZF129 not only showed the effectiveness on the control of cucumber root rot, but also has a growth promoting effect on cucumber plants. Future research will continue to focus on improving the storage stability of biocontrol bacteria in different environments and expanding the application of this formulation in other plant disease control.”
Comments 33: Many figures and tables lack clear labels, annotations, or units (e.g., Figure 2, Table 1).
Response 33: Thank you for pointing this out. We have checked and modified the labels, annotations, or units in the charts and tables in the article.
Comments 34: Error bars or standard deviations are missing in some figures (e.g., Figure 5).
Response 34: Thank you for pointing this out. The missing standard deviation in Figure 5 is due to the fact that all three values are 0, so it cannot be annotated
Comments 35: Key terms (e.g., "colonization ability," "control efficacy") are used inconsistently or without clear definitions.
Response 35: Thank you for pointing this out. Key terms were consistently revised in the updated manuscript.
Comments 36: The use of rifampicin-resistant strains is not discussed in the context of potential risks (e.g., antibiotic resistance gene transfer).
Response 36: Rifampicin-resistant strains were only used in the lab and will not be used in the field. The bacteria, plants and soil involved in the test were sterilized. So there is no risks.

Round 2
Reviewer 1 Report
Comments and Suggestions for Authors
Thanks for the detailed and thoughtful responses from the authors. Your revisions have significantly improved the manuscript and have addressed my previous concerns.
However, there are still some issues that need to be refined. Please see my notes below for further improvements, especially those marked in red font.
- Lines 32-43 on pages 1-2 in the Abstract section, these lines will be removed since Lines 19-31 on page 1 is the revised one. Do I understand right?
- The newly inserted sentences on Lines 58–64 are not well connected with the surrounding context, and there is a grammatical mistake in this part. Please correct and revise them for better coherence. Similar issues occur in the newly inserted sentences on Lines 80–87. Therefore, I suggest reviewing the entire manuscript.
- For Figure 1, the significant markers (*) typically indicate differences between different treatments at each time point rather than representing the curve for each treatment.
- In Figure 3, which images are before coating and which are after? Please clarify.
- In Figure 5, the figure includes error bars, but there is no indication of whether they represent S.D. or S.E.M. Additionally, there is no note clarifying the different letters used to indicate statistical significance.
- It is odd to suddenly introduce the background of these four types of soil enzymes. Their information could be mentioned in the discussion alongside the results in Figure 8, analyzing what the results indicate, rather than abruptly presenting this information without connection to the context.
- On Lines 442–443, page 12, it is incorrect to state that 'the ZF129 did not have a significant impact on the activity of the other three soil enzymes.' According to Figure 8, there is a significant impact, though not a significant improvement.
- In Figure 9a, the significant markers are incorrectly labeled, leading to no significant differences among the four, which is not consistent with the content on Lines 448–453 on page 12. Please correct this.
- Figure 9c shows that T4 is significant different with other three in SOD enzyme activity, which is not consistent with the content on Lines 452–453 on page 12.Please correct this.

Please pay close attention and double-check the coherence of the entire manuscript. For example, the newly inserted sentences on Lines 58–64 are not well connected with the surrounding context, and there is a grammatical mistake in this section. Please correct and revise them for better coherence. Similar issues occur in the newly inserted sentences on Lines 80–87. Therefore, I suggest reviewing the entire manuscript.
Author Response
Referee: 1
Thanks for the detailed and thoughtful responses from the authors. Your revisions have significantly improved the manuscript and have addressed my previous concerns.
Response: We sincerely thank the reviewer for the reminder. We have revised the manuscript according to the suggestions of Reviewers. We hope that the revised manuscript will be published finally and make our contribution in the research field of control of cucumber Fusarium root rot.
Comments 1: Lines 32-43 on pages 1-2 in the Abstract section, these lines will be removed since Lines 19-31 on page 1 is the revised one. Do I understand right?
Response 1: We thank the reviewer for the helpful comments. You understand completely correctly. I'm very sorry that the previous content was not deleted after the revision. It has now been revised.
Comments 2: The newly inserted sentences on Lines 58–64 are not well connected with the surrounding context, and there is a grammatical mistake in this part. Please correct and revise them for better coherence. Similar issues occur in the newly inserted sentences on Lines 80–87. Therefore, I suggest reviewing the entire manuscript.
Response 2: Thank the reviewer for the grammar check. We have made modifications to this grammar section to maintain the fluency of the context, and have checked the entire manuscript.
Line 46-55:
While chemical pesticides remain a conventional approach for disease manage-ment, their indiscriminate application has prompted critical reassessment due to esca-lating operational expenditures and ecological contamination risks. Concurrently, prolonged pesticide utilization accelerates pathogen resistance development through both target-site modification and detoxification enhancement mechanisms, systemati-cally diminishing the long-term viability of current control paradigms [4]. The fungi-cides, propiconazole and difenoconazole, and premixtures, pydiflumetofen + fludioxo-nil or pydiflumetofen + difenoconazole, provided excellent root decay control. Azoxystrobin provided excellent (69.9%) control of Phoma root decay of table beet in 2019 and lesser (40%) control in 2021 [5].
Line 71-78:
Colla et al. (2015) coated wheat seeds with endophytic fungi, increasing stem, root biomass, and leaf number of winter wheat seedlings by 23%, 64%, and 29%, respectively. Yield increased by 8.3% to 32.1%, grain protein concentration increased by 6.3%, and grain potassium, phosphorus, iron, and zinc concentrations also generally increased [8]. Angelopoulou et al. (2014) found that the use of two seed coating agents, Bacillus subtilis K165 and non pathogenic Fusarium oxysporum F2, delayed the onset of eggplant wilt disease and significantly reduced the disease index, and also promoted plant growth [9].
Comments 3: For Figure 1, the significant markers (*) typically indicate differences between different treatments at each time point rather than representing the curve for each treatment.
Response 3: We speculate that you may be referring to Figure 2 instead of Figure 1 when comparing the entire manuscript. According to the comments of another reviewer, important markers have been modified, indicating that microencapsulation can significantly improve the survival ability of the P. polymyxa ZF129 in the five weeks. We believe that the differences between different treatments are already very significant, and there is no need to annotate the importance of the differences in the same treatment at different times.
Comments 4: In Figure 3, which images are before coating and which are after? Please clarify.
Response 4: We thank the reviewer for the comments. We have already explained in the image annotation that a, b are cucumber seeds before coating, and c, d are cucumber seeds after coating (Line 302-303).
Comments 5: In Figure 5, the figure includes error bars, but there is no indication of whether they represent S.D. or S.E.M. Additionally, there is no note clarifying the different letters used to indicate statistical significance.
Response 5: We sincerely appreciate the reviewer's reminder on the explanation of standard deviation. We have marked the error bars as standard deviation (S.D.). In addition, annotations were provided for different letters indicating statistical significance (Line 407-411).
Comments 6: It is odd to suddenly introduce the background of these four types of soil enzymes. Their information could be mentioned in the discussion alongside the results in Figure 8, analyzing what the results indicate, rather than abruptly presenting this information without connection to the context.
Response 6: We sincerely thank the reviewer for the helpful suggestions We have revised this section by incorporating this information with the results in Figure 8 and mentioning it in the discussion section, as indicated by the analysis results. We also removed the relevant explanations for the other three soil enzymes (S-SC, S-AKP/ALP, and S-CAT) whose activity did not increase (Line 423-431).
Line 442-430:
There were higher S-UE and S-SC activities in soil infected with F. solani. Com-pared with the F. solani control (T4), coating seed with ZF129 significantly improved the S-UE activity by 38.82% (T1) and 13.92% (T2). S-UE can catalyze the hydrolysis of urea into ammonia (NH3) and carbon dioxide, and the released ammonia is converted into nitrate (NO3 ⁻) through nitrification, providing the main nitrogen source for plants (accounting for 40-60% of their nitrogen requirements. However, the ZF129 did not have a significant impact on the activity of the other three soil enzymes (S-SC, S-AKP/ALP, and S-CAT). This indicates that ZF129 does not play a role in promoting the secretion of plant growth factors or stimulating microbial activity.
Comments 7: On Lines 442–443, page 12, it is incorrect to state that 'the ZF129 did not have a significant impact on the activity of the other three soil enzymes.' According to Figure 8, there is a significant impact, though not a significant improvement.
Response 7: We thank the reviewer for the helpful comments. We have optimized the statement of 'the ZF129 did not have a significant impact on the activity of the other three soil enzymes' (Line 428-429).
Comments 8: In Figure 9a, the significant markers are incorrectly labeled, leading to no significant differences among the four, which is not consistent with the content on Lines 448–453 on page 12. Please correct this.
Response 8: We thank the reviewer for the reminder. We have corrected the annotation error of the significant markers in Figure 9 a.
Comments 9: Figure 9c shows that T4 is significant different with other three in SOD enzyme activity, which is not consistent with the content on Lines 452–453 on page 12.Please correct this.
Response 9: We thank the reviewer for the reminder. Thank you for pointing this out. We have corrected the annotation error of the significant markers in Figure 9 c.
Round 3
Reviewer 1 Report
Comments and Suggestions for Authors
Thanks for the authors' reply. There are several minor issues I am concerned.
- I still insist on that the significant markers (*) typically indicate differences between different treatments at each time point rather than representing the curve for each treatment. The authors should revise Figure 2 to conform to the formatting standards.
- In Figure 3, it is clearly explained in the image annotation that a, b are cucumber
seeds before coating, and c, d are cucumber seeds after coating (Line
302-303). What I am confused is that the difference between a and b. Are they two replicates? what about c and d. Are they two replicates, too? Please describe it clearly in the figure legend.
Author Response
Referee: 1
Thanks for the authors' reply. There are several minor issues I am concerned.
Response: We sincerely thank the reviewer for the helpful comments. We have revised the manuscript according to the suggestions of Reviewers. We hope that the revised manuscript will be published finally and make our contribution in the research field of control of cucumber Fusarium root rot.
Comments 1: I still insist on that the significant markers (*) typically indicate differences between different treatments at each time point rather than representing the curve for each treatment. The authors should revise Figure 2 to conform to the formatting standards.
Response 1: We thank the reviewer for the comments. We have made modifications to Figure 2 by using significant markers (*) to represent the differences between different treatments at each time point.
Comments 2: In Figure 3, it is clearly explained in the image annotation that a, b are cucumber seeds before coating, and c, d are cucumber seeds after coating (Line 302-303). What I am confused is that the difference between a and b. Are they two replicates? what about c and d. Are they two replicates, too? Please describe it clearly in the figure legend.
Response 2: Thank the reviewer for the Figure 3 check. We have provided additional descriptions for Figure 3: (a, b) Cucumber seeds before coating at different magnifications; (c, d) Cucumber seeds after coating at different magnifications.